# Near-Infrared Spectroscopy (NIRS) versus Hyperspectral Imaging (HSI) to Detect Flap Failure in Reconstructive Surgery: A Systematic Review

**DOI:** 10.3390/life12010065

**Published:** 2022-01-03

**Authors:** Anouk A. M. A. Lindelauf, Alexander G. Saelmans, Sander M. J. van Kuijk, René R. W. J. van der Hulst, Rutger M. Schols

**Affiliations:** 1Department of Cardiothoracic Surgery, Maastricht University Medical Center, Debyelaan 25, P.O. Box 5800, 6202 AZ Maastricht, The Netherlands; a.g.saelmans@gmail.com; 2Department of Plastic, Reconstructive and Hand Surgery, Maastricht University Medical Center, 6229 HX Maastricht, The Netherlands; r.vander.hulst@mumc.nl; 3Department of Clinical Epidemiology and Medical Technology Assessment, Maastricht University Medical Center, 6229 HX Maastricht, The Netherlands; sander.van.kuijk@mumc.nl

**Keywords:** free flap, near-infrared spectroscopy, hyperspectral imaging, flap failure, flap loss, tissue oxygenation, non-invasive monitoring

## Abstract

Rapid identification of possible vascular compromise in free flap reconstruction to minimize time to reoperation improves achieving free flap salvage. Subjective clinical assessment, often complemented with handheld Doppler, is the golden standard for flap monitoring; but this lacks consistency and may be variable. Non-invasive optical methods such as near-infrared spectroscopy (NIRS) and hyperspectral imaging (HSI) could facilitate objective flap monitoring. A systematic review was conducted to compare NIRS with HSI in detecting vascular compromise in reconstructive flap surgery as compared to standard monitoring. A literature search was performed using PubMed and Embase scientific database in August 2021. Studies were selected by two independent reviewers. Sixteen NIRS and five HSI studies were included. In total, 3662 flap procedures were carried out in 1970 patients using NIRS. Simultaneously; 90 flaps were performed in 90 patients using HSI. HSI and NIRS flap survival were 92.5% (95% CI: 83.3–96.8) and 99.2% (95% CI: 97.8–99.7). Statistically significant differences were observed in flap survival (*p* = 0.02); flaps returned to OR (*p* = 0.04); salvage rate (*p* < 0.01) and partial flap loss rate (*p* < 0.01). However, no statistically significant difference was observed concerning flaps with vascular crisis (*p* = 0.39). NIRS and HSI have proven to be reliable; accurate and user-friendly monitoring methods. However, based on the currently available literature, no firm conclusions can be drawn concerning non-invasive monitoring technique superiority

## 1. Introduction

One of the most feared complications in reconstructive flap surgery is flap failure as a consequence of microvascular thrombosis. Usually, vascular compromise occurs within 48 h after surgery [1,2]. Achieving free flap salvage is improved by rapid identification of possible complications to minimize time to reoperation [3]. In theory, the ideal method of monitoring would be continuous, non-invasive, sensitive enough to detect vascular compromise instantly, sufficiently reliable to make specialized nursing care dispensable, easy to use, harmless to the patient and flap, applicable to all types of flaps, and inexpensive [4,5,6,7].

Monitoring traditionally consists of the subjective assessment of skin color, capillary refill time, temperature and tissue turgor. Frequently, techniques such as handheld Doppler ultrasound, implantable Doppler probes, temperature probes and color duplex sonography are used in conjunction. However, differences in level of clinical experience in free flap monitoring of medical staff influences the consistency of recordings and increases variability. Additionally, these methods are labour intensive, performed intermittently, and one is not clearly superior to another [8,9,10,11,12]. Therefore, more objective methods are desired for flap monitoring. 

Near-infrared spectroscopy (NIRS) is a non-invasive continuous bedside monitoring technique of flap tissue oxygenation that could potentially live up to this demand. Selective absorption of near-infrared light during transmission through the tissue by oxygen-dependent chromophores (hemoglobin) is measured. The percentage of saturated hemoglobin (StO_2_) is calculated based on the ratio of oxygenated (HbO_2_) and deoxygenated (Hb) hemoglobin. StO_2_ is associated with tissue oxygenation determined by the balance between oxygen delivery and consumption. Therefore, it indirectly reflects the status of tissue perfusion [7,11,13,14]. NIRS can be used to monitor buried flaps as long as the thickness of the overlying skin does not exceed the maximum depth range of the sensor used [11]. However, measuring tissue oxygenation intraoperatively is currently not possible, since sterile sensors are not available.

Another promising method that can be applied to assess the quality of tissue perfusion is hyperspectral imaging (HSI). HSI is a non-invasive, contactless monitoring technique that combines the principles of imaging and spectroscopy. The technique processes the optical properties of the flap area in a wavelength spectrum from visual to near-infrared light. Consequently, a three-dimensional data set is acquired. HSI provides objective, precise, reproducible and relevant information about 4 parameters in tissue perfusion measurements. The cutaneous and subcutaneous oxygenation patterns are analyzed with hemoglobin oxygenation (StO_2_) and Near-infrared Perfusion index (NPI or NIR (PI)), measuring the superficial hemoglobin oxygen saturation with a penetration depth of consecutively 1 mm and of 3–5 mm. Tissue Hemoglobin Index (THI) displays the distribution of hemoglobin in the flap microcirculation. Tissue Water Index (TWI) provides information concerning water content and distribution in the flap [15,16,17,18,19,20]. Despite the measurement not being continuous, this monitoring technique enables the assessment of flap viability intraoperatively.

This review aims to compare NIRS with HSI in detecting vascular compromise in reconstructive flap surgery compared to standard monitoring. 

## 2. Materials and Methods

This literature review is reported in accordance with the Preferred Reporting Items for Systematic Reviews and Meta-Analyses (PRISMA) guideline. The PRISMA checklist is provided in Appendix B. The review was registered prospectively on Prospero (receipt number 274,196; formal approval is pending). A systematic literature search was performed by two reviewers independently (AL/AS) utilizing the National library of medicine (PubMed) database and Embase scientific database (via OvidSP). The literature search was completed in August 2021. The search was performed separately for both databases. Various medical subject heading (MeSH) terms combined with free search terms were used as depicted in Table 1. Studies conducted other than in humans, reviews and studies published in languages other than Dutch, English, German, French and Spanish were excluded from this review. A detailed search query is provided in Appendix B. For the selection of the studies included in this study the Population, Intervention, Comparison, Outcome and Study Design (PICOS) approach was used. After removal of the duplicates, eligibility of the remaining articles was primarily determined by screening based on title. Subsequently, studies were screened based on the abstract. Remaining studies were screened by reading the full text; those that did not answer the research question of this review were excluded. In case of disagreement between the two reviewers AL/AS a third researcher (RS) was consulted. 

### 2.1. Data Extraction

From the included studies, the following information was extracted: the surname of the first author, country of origin, year of publication, study design, study period, researched monitoring tool, monitoring protocol, study objective, number of patients, number of flaps, age, sex, Body Mass Index (BMI), flap survival, monitoring control technique, bilateral flaps, flap weight, mean ischemia time, types of flaps, vascular disease, diabetes mellitus, smoking, radiotherapy, chemotherapy, prior abdominal surgery, use of inotropes, decisive monitoring tool, warning value, flaps with vascular crisis, flaps returned to OR, salvage rate, average time to discharge, total flap loss rate, partial flap loss rate, sensitivity and specificity.

### 2.2. Data Synthesis

Systematic review methodology and standard summary statistics overall were used to summarize available evidence. Study-level data was analyzed using meta-regression using a random-effects model. The analysis was performed in R 4.1.1 (R Foundation for Statistical Computing, Vienna, Austria) with the ‘meta’ package. Meta-regression was carried out for the following outcomes: flap survival, flaps with vascular crisis, flaps returned to OR, salvage rate and partial flap loss. Because of significant methodological and statistical heterogeneity between the included studies, further meta-analytic methods were not applied.

## 3. Results

### 3.1. Literature Search

In total, twenty one of the 428 studies that were found with our search strategy qualified for inclusion in this study, see Figure 1. All twenty-one studies were single center studies, except one. Eleven studies were performed in Europe, two in Asia, and eight in the USA during the period from January 2004 to January 2020. Sixteen studies reported on the use of NIRS to detect flap failure and five studies reported on the use of HSI to prevent flap failure. Twenty studies had a cohort study design. Most of the included studies had a prospective design; ten of the NIRS and three of the HSI studies. Seven studies reported retrospectively collected data: six of the NIRS and one of the HSI studies. One HSI study was a case report. No randomized trials were identified. According to the ROBINS-I, the risk of bias assessment of the observational studies is presented in Figure 2 and Figure A1. The bias assessment of one case report was carried out with the Newcastle-Ottawa Scale [20]. Among these studies, inclusion criteria were comparable (Table A1)

### 3.2. Overall Flap Surgery Patient Profiles

Data of 2686 patients extracted from 21 studies who underwent flap surgery and were consequently monitored were analyzed (Table 2). The flaps were monitored with either NIRS in 1970 (73.3%), HSI in 90 (3.4%) or standard monitoring alone in 626 (23.3%) patients. The devices used for NIRS monitoring were ViOptix in eight (T.Ox 6, ODIsey 1), Inspectra in three (M325 1, M650 2), INVOS in two (5000C 1, 5100C 1), TSNIR-3 in one and TOS-96/TOS-OR in one study. For HSI TIVITA was used in four and ImSpector V8E in one study. The control monitoring technique consisted of clinical examination in twenty and indocyanine green (ICG) fluorescence [21] imaging in one. Control monitoring was carried out in conjunction with Doppler in eleven studies and with ICG imaging in one. The average/median ages in the included studies are depicted in Table 2. Females accounted for 91.8% (1861/2027) overall; 19.3% (11/57) of the HSI and 93.9% (1850/1970) of the NIRS study population. Gender data weren’t described in one HSI study [22], age wasn’t described in two studies [22,23]. Data on body mass index (BMI) was reported by nine NIRS studies. 3662 flaps were monitored with NIRS in 2759 (75.3%), HSI in 90 (2.5%) or standard monitoring alone in 813 (22.2%) flaps. The overall flap survival was 98.8% (95% CI: 97.1–99.5); HSI and NIRS flap survival respectively were 92.5% (95% CI: 83.3–96.8) and 99.2% (95% CI: 97.8–99.7). This difference was statistically significant (*p* = 0.02).

### 3.3. Flap-Related Characteristics

Data depicting flap related characteristics were reported inconsistently, except for types of flaps. Therefore, substantial amount of data was not available. Flap types, ischemia time, vascular disease, Diabetes Mellitus, smoking, radiotherapy and chemotherapy are described in Table 3. In one study 14 (47%) patients received (neo)-adjuvant therapy prior to surgery consisting of immunotherapy, endocrine therapy, radiation therapy, chemotherapy or a combination of these [24]. Prior abdominal surgery is described in two studies: 52 (26.5%) [25] and 214 (56.5%) in the control group, 356 (53.1%) in the NIRS group [26]. None of the included studies described use of inotropes.

### 3.4. Detection of Flap Failure

In at least nine out of sixteen studies, NIRS was the first tool indicating flap failure. Data regarding the first monitoring tool to detect complication was not provided by five studies. A faster detection with standard monitoring was observed in one study and with ICG imaging or standard monitoring in another compared with NIRS [27,28]. HSI was the first tool to indicate a vascular crisis in at least two out of five studies [22,29]; the other three didn’t provide data concerning the first tool to detect flap complication (Table 4). Time to detection was not mentioned in any of the included studies. The cut-off value for detection of flap complication was mentioned in the majority of studies. Proposed warning values for specific flap monitoring models according to recent studies and parameters to distinguish venous congestion from arterial occlusion are indicated in Table 5. Overall, 6.0% (95% CI: 4.0–8.9) of flaps had vascular crisis. Flaps monitored using HSI presented with vascular crisis in 10.0% (95% CI: 5.3–18.1) and using NIRS in 5.5% (95% CI: 3.4–8.8). This difference was not statistically significant (*p* = 0.39). Overall, 6.3% (95% CI: 4.3–9.1) of flaps were returned to the OR. In the HSI studies 12.7% (95% CI: 5.5–26.7) and in the NIRS studies 5.6% (95% CI: 3.8–8.2) of flaps were returned to the OR. This difference was statistically significant (*p* = 0.04). Salvage rate, the percentage of flaps with vascular crisis that could be saved, overall was 81.1% (95% CI: 65.1–90.8). HSI salvage rate was 22.2% (95% CI: 5.6–57.9) and NIRS salvage rate was 88.3% (95% CI: 80.1–93.4). This difference was statistically significant (*p* < 0.01). Average time to discharge was mentioned in 6 studies and is depicted in Table 4. Partial loss rate overall was 0.57% (95% CI: 0.13–2.52), for HSI 6.64% (0.44–53.36) and for NIRS 0.60% (95% CI: 0.19–1.89). This difference was statistically significant (*p* < 0.01). Sensitivity and Specificity were described in 13 studies.

**Table 2 life-12-00065-t002:** Baseline characteristics of included population (n = 2686) and outcomes.

Author	Country	Year of Publication	Study Type	Study Period	Researched Monitoring Tool (model)	Objective	Patients (N)	Flaps (N)	Age (years)	Female (N, %)	Body Mass Index (kg/m^2^)	Flap Survival (%)	Monitoring Control Technique
**Cai [30]**	China	2007	Prospective	Na	NIRS (TSNIR-3) first 24 h every 4 h, 2 per day in the following 5 days	Test sensibility and precision	41	41	42 (14–73)	11 (26.8)	Na	97.6	CE
**Carruthers [25]**	USA	2019	Retrospective	24 months	NIRS (T.Ox ViOptix) until discharge	Reduce monitoring time	196	301	50.7 ± 8.3	196 (100)	30.7 ± 5.5	100	CE, pD every hour first 12 h, every 4 h
**Guye [31]**	France	2017	Prospective	7 months	NIRS (InSpectra Model 650)	Reassess risk factors for free flap complications	40	40	NC 53.3 ± 13.6, C 58 ± 15.9	18 (45)	C 26 ± 1.7, NC 22 ± 0.7	90	CE
**Keller [23]**	USA	2009	Prospective	Jan 2005– Jan 2008	NIRS (T.Ox ViOptix) for 36 h	Continuation of an earlier preliminary study	145	208	Na	145 (100)	Na	100	CE, hD; hourly for first night, every 2 h for the next 36 h
**Koolen [26]**	USA	2016	Retrospective	Feb 2004–Jun 2008, Jun 2008–Feb 2014	Control, NIRS (T.Ox ViOptix) for 72 h	Compare standard monitoring with NIRS	Co 288, Ni 451	Co 380, Ni 670	Co 47.7 ± 7.9, Ni 49.9 ± 8.5	Co 288 (100), Ni 451 (100)	Co 26.9 ± 5, Ni 28.9 ± 5.6	Co 57.7, Ni 96.6	CE
**Lin [32]**	USA	2010	Retrospective	Jan 2004–Dec 2007, Jan 2008–Dec 2010	Control, NIRS (T.Ox ViOptix) for 72 h	Compare monitoring with and without tissue oximetry	Co 288, Ni 164	Co 380, Ni 234	Co 47.69 ± 8.44, Ni 49.85 ± 7.88	Co 288 (100), Ni 164 (100)	Na	Co 97.1, Ni 99.6	CE, hD; every 15min first hr, every 30 min second hr, every hour for next 10 h. Surgical resident every 4 h CE.
**Lindelauf [24]**	NL	2021	Prospective	Na	NIRS (FORE-SIGHT MC-2030) for 24 h	Confirm the usefulness of NIRS in postoperative monitoring	30	42	51 ± 13	30 (100)	27.5 ± 4.3	100	CE, Doppler according to hospital protocol
**Lohman [8]**	USA	2013	Prospective	Jan 2006–Feb 2007	NIRS (ViOptix) for 72 h	Determine the most useful method	38	38	38.5 (21–84)	27 (71.1)	Na	100	CE, hD, hourly;iD
**Pelletier [33]**	USA	2011	Prospective	Aug 2006–Jan 2010	NIRS (ViOptix ODIsey) for 72 h	Evaluate the costs of autologous free tissue breast reconstruction	50; ICU 25, Floor 25	54	ICU 49.4 (31–67), Floor 49 (28–75)	50 (100)	ICU 27.9 (19.5–43), Floor 28.5 (21.8–36.3)	98	CE, hD; ICU every hour, Floor every 4–6 h
**Repez [11]**	Slovenia	2007	Prospective	Aug 2004–Sep 2005	NIRS (InSpectra Model 325) for 72 h	Ascertain whether NIRS could be trustworthy	48	50	47 (31–64)	48 (100)	26 (22–35)	94	CE hourly for 72 h
**Ricci [34]**	USA	2017	Retrospective	May 2008–Aug 2014	NIRS (ViOptix T.Ox) for 72 h	Earlier transfer of patients to a standard surgical inpatient floor	595	900	50.3 ± 8.6	595 (100)	28.8 ± 5.6	99.7	CE first 24 h, hD; every 15 min for first hr, every 30 min for second hr, every hr for the next 22 h
**Salgarello [35]**	Italy	2018	Retrospective	Jan 2015–Jan 2016	NIRS (INVOS 5100C) for 48 h	Identify patient- and flap related variables that can affect rSO_2_	45	45	52.6 (34–69)	45 (100)	**	100	ICG imaging
**Steele [36]**	USA	2011	Retrospective	Jan 2007–May 2010	Control, NIRS (ViOptix T.Ox) for 4.5d average	Examine outcomes using a tissue oximeter	Co 50, Ni 63	Co 53, Ni 75	Co 57.6 (11–85), Ni 58 (17–89)	Co 18 (36), Ni 29 (46)	Na	Co 90.6, Ni 98.7	CE, hD, hourly for 48 h, then every 2 h for the following 48 h, then every 4 h *
**Vranken [27]**	NL	2017	Prospective	Na	NIRS (INVOS 5000C) for 24 h	Suitability for the assessment of tissue perfusion	29	29	50 ± 10	29 (100)	26.4 ± 3.3	100	CE, Doppler ultrasonography
**Whitaker [37]**	UK	2012	Prospective	Na	NIRS (InSpectra Model 650) for 72 h	Investigate NIRS technology	10	10	46 (28–61)	10 (100)	Na	90	CE, hD (hourly), capillary bleeding (25 gauge needle)
**Yano [28]**	Japan	2020	Prospective	Sep 2011–Jan 2016	NIRS (TOS-96/TOS-OR) for 72 h	Investigate the feasibility of perioperative NIRS monitoring	25	25	63.5 (39–85)	2 (8)	Na	100	CE, ICG imaging
**Calin [20]**	Romania	2017	Case report	Na	HSI (ImSpector V8E) 0, 2, 4, 24 and 48 h postoperatively	Assess value as a monitoring tool	1	1	61	0	Na	100	CE
**Kohler [29]**	Germany	2021	Prospective	Mar 2019–Jan 2020	HSI (TIVITA) at t0(0), t1(16–28), t2(39–77) hrs postoperatively	Show the superiority of HSI	22	22	55 (26–92)	5 (22.7)	Na	81.8	CE, Doppler ultrasound every 2 h within 24 h, every 4 h until 72 h postoperatively
**Schulz (‘20) [38]**	Germany	2020	Retrospective	Dec 2017–Apr 2018	HSI (TIVITA) for 7 days	Evaluate HSI as a monitoring method for pedicled flaps	16	16	58 (25–78)	2 (12.5)	Na	93.8	Na
**Schulz (‘21) [39]**	Germany	2021	Prospective	Jul 2017–Sep 2018	HSI (TIVITA) for 7 days	Investigate HSI as a method for free flap monitoring	18	18	54 (24–87)	4 (22.2)	Na	94.4	CE
**Thiem [22]**	Germany	2020	Prospective	Na	HSI (TIVITA) at t1(0), t2(0–1), t3(4–8), t4(8–12), t5(12–24), t6(24–48), t7(>48)	Feasibility of HSI for objective and reproducible monitoring	33	33	Na	Na	Na	97	CE 72 h

Co = control, Ni = NIRS, C = complication, NC = No complication, CE = clinical examination, hD = handheld Doppler, pD = pencil Doppler, iD = implantable Doppler, Na = not available, * Implantable Doppler was used in a few patients whose flaps were completely buried, ** Salgarello et al. BMI 18.5–24.9; N = 24, BMI 25–29.9; N = 11, BMI > 30; N = 10.

**Table 3 life-12-00065-t003:** Flap-related characteristics of included flaps (n = 3662).

Author	Mean Ischaemia Time (min)	Types of Flaps (N)	Vascular Disease (N, %)	DM (N, %)	Smoker (N, %)	XRT (N, %)	Chemo (N, %)
**Cai [30]**	Na	Fibular 41	Na	Na	Na	Na	Na
**Carruthers [25]**	Na	DIEP 301 (111 delayed, 36.9%)	Na	8 (4.1)	9 (4.6)	78 (25.9)	Na
**Guye [31]**	NC 74 ± 4.5, C 70 ± 6.8	Fibular 15, Radial 20, gastro-omental 5	5 (12.5)	5 (12.5)	Na	9 (22.5)	Na
**Keller [23]**	Na	DIEP 197, SIEA 1, SGAP 10	Na	Na	Na	Na	Na
**Koolen [26]**	Na	Co; DIEP 336, SIEA 15, Free TRAM 9, SGAP 20 Ni; DIEP 646, SIEA 1, Free TRAM 3, SGAP 20	CAD Co 1 (0.3), Ni 5 (0.7)	Co 8 (2.1), Ni 28 (4.2)	Co 30 (7.9), Ni 85 (12.7)	Co 105 (27.6), Ni 235 (35.1)	Co 157 (41.4), Ni 379 (58.2)
**Lin [32]**	Na	Co; DIEP 336, SIEA 15, SGAP 20, Free TRAM 9 Ni; DIEP 222, SGAP 9, Free TRAM 3	Na	Na	Na	Na	Na
**Lindelauf [24]**	42 (35–51)	DIEP 42 (17 secondary)	Na	Na	2 (7)	Na	Na
**Lohman [8]**	Na	DIEP 18, ALT 15, MS-TRAM 5	Na	Na	Na	Na	Na
**Pelletier [33]**	ICU 86.7 (46–157), Floor 78.5 (48–138)	DIEP 21, DIEP + DIEP 1, DIEP/SIEV 2, DIEP + SIEA 3, SIEA 9, Free TRAM 3, Free MS-TRAM 11	0	Floor 1, ICU 0	Floor 1, ICU 0	Floor 12, ICU 12	Floor 11, ICU 12
**Repez [11]**	Na	DIEP 37 (13 secondary), SIEA 5, SGAP 8 (5 secondary)	0	1 (2)	7 (14)	Na	Na
**Ricci [34]**	Na	DIEP 872, SIEA 2, SGAP 23, TRAM 3	CAD 5 (<0.1)	32 (3.6)	89 (9.9)	265 (29.4)	414 (46)
**Salgarello [35]**	Na	DIEP 45	Na	Na	Na	Na	Na
**Steele [36]**	Na	Co; DIEP 5, ALT 7, Fibular 5, LD 3, Scapula osteocutaneous 1, Free TRAM 14, Radial 14, gracilis 4 Ni; DIEP 26, ALT 20, Fibular 8, Free TRAM 2, MS-TRAM 4, Radial 15	Na	Na	Na	Na	Na
**Vranken [27]**	48 ± 12	DIEP 29	Na	Na	Na	Na	Na
**Whitaker [37]**	Na	DIEP 10	Na	Na	Na	Na	Na
**Yano [28]**	Na	FJG 25	Na	Na	Na	6 (24)	20 (80)
**Calin [20]**	Na	Fasciocutaneous sural flap 1	1 (100)	1 (100)	Na	Na	Na
**Kohler [29]**	Na	DIEP 3, ALT 11, LD 4, Scapula osteocutaneous 1 (parascapular), MS2-TRAM 2, Rectus abdominis 1 (18 with, 4 without skin island)	PAD 4 (18.2), CAD 3 (13.6)	5 (22.7)	4 (18.2)	Na	Na
**Schulz (‘20) [38]**	Na	Suralis 3, LD 5, Radial 2, gastrocnemius 2, TFL 1, Foucher 1, MCPA 1, Crossfinger 1	Na	Na	Na	Na	Na
**Schulz (‘21) [39]**	Na	ALT 10, LD 8	PAD 7 (38.9), CAD 5 (27.8)	8 (44.4)	12 (66.7)	Na	Na
**Thiem [22]**	Na	*25 FF*; ALT 3, Radial 12, Osteocutaneous fibula 4, Osteocutaneous scapular 3, Unknown 3 *8 PF*; PM 3, LD 2, NL 1, LSS 2	Na	Na	Na	Na	Na

Co = control, Ni = NIRS, C = complication, NC = no complication, DIEP = Deep inferior epigastric artery perforator, ALT = Anterolateral thigh, SIEA = Superficial inferior epigastric artery, SGAP = superior gluteal artery perforator, LD = Latissimus Dorsi, TRAM = Transverse rectus abdominis myocutaneous, MS2 = Muscle sparing type 2, TFL = Tensor fascia lata, MCPA = metacarpal arteries, FJG = Free jejunal graft, PM = Pectoralis major, NL = Nasolabial, LSS = Large scale scalp rotation, PAD = peripheral artery disease, CAD = coronary artery disease, DM = Diabetes Mellitus, XRT = radiation therapy, FF = Free flap, PF = Pedicled flap, Na = not available.

**Table 4 life-12-00065-t004:** Detection of flap complication.

Author	Decisive Monitoring	Warning Value	Description	Flaps with Vascular Crisis (N, %)	Flaps Returned to OR (N, %)	Salvage Rate (%)	Average Time to Discharge (days)	Total Loss Rate (N, %)	Partial Loss Rate (N, %)	Sensitivity (%)	Specificity (%)
**Cai [30]**	NIRS	rSO_2_ 70%	Anastomosis vein again, intraoral infection day 7, necrosis 1	venous 1 (2.4)	1 (2.4)	0	Na	1 (2.4)	0	100	100
**Carruthers [25]**	NIRS (5 microvascular)	rSO_2_	Microvascular 5 (3 immediate reconstructions, 2 delayed cases), Nonvascular 9 (1 positive margin required reexcision, 8 hematoma)	5 (1.7); venous congestion 3, arterial thrombus 2	14 (4.7)	100	3.4 ± 1.1	0	0	Na	Na
**Guye [31]**	Na	Na	Venous thrombosis 2, Partial or total necrosis of the flap 8 ( arterial thrombosis 3)	venous thrombosis 2 (5)	Na	Na	Na	4 (10)	4 (10)	Na	Na
**Keller [23]**	NIRS	ΔStO_2_/Δt ≥ 20%/h sustained >30 min	Hematoma, superficial vein thrombosis and vein kink 1, Deep vein thrombosis 2, Arterial thrombosis 2	5 (2.4); venous 3, arterial 2	5 (2.4, 1 triple)	100	Na	0	0	100	100
**Koolen [26]**	Na	Co Na, Ni 20-point drop in 1 h OR absolute reading <30%	Na	Na	Co 26 (6.8), Ni 29 (4.3)	Co 57.7, Ni 96.6	Na	Co 11(2.9), Ni 1(0.1)	Co 8 (2.1) Ni 7 (1)	Co Na, Ni 96.5	Co Na, Ni 99.8
**Lin [32]**	Co CE/hD, Ni NIRS	Co Na, Ni 20-point drop in 1 h OR absolute reading <30%	Co; Na, Ni; Venous thrombosis resulted in total loss 1	Co 26 (6.8), Ni 16 (6.8)	Co 26 (6.8), Ni 16 (6.8)	Co 57.7, Ni 93.8	Na	Co 11 (2.9), Ni 1 (0.43)	Co 8 (2.1), Ni 4 (1.7)	Co Na, Ni 100	Co Na, Ni 100
**Lindelauf [24]**	Na	Na	Minor complication 13, Major complication 5 (debridement for fat necrosis 1, arterial kinking 1, evacuation hematoma 1, insufficient perfusion resulting in partial loss 1, venous kinking 1)	3 (7.1)	5 (12)	100	no/minor c 5 [4,5]	0	1 (2.4)	100	100
**Lohman [8]**	NIRS ( in ⅘)	StO_2_ ≤30%	Hematomas 2, Venous thrombosis 1, Venous kinking and clotting 1, Venous clotting 1	4 (13.2); 3 venous, 1 arteriovenous	5 (13.2)	100	Na	0	0	100	100
**Pelletier [33]**	NIRS	StO_2_ <30% OR StO_2_ >20%/h drop for 30 min	Venous thrombosis 3, No reoperation 1	4 (8); ICU venous 3, Floor 1	3 (6)	75	Na	ICU 1 (2)	0	100	100
**Repez [11]**	NIRS	StO_2_ <50% of initial value	Venous thrombosis 8, Arterial thrombosis 2	10 (20); arterial 2, venous 11	10 (20, 1 twice, 1 triple)	70	Na	3 (6)	0	100	100
**Ricci [34]**	Na	20-point drop in 1 h OR absolute reading <30%	Venous thrombosis, pedicle kinking or hematoma causing compression 25, Arterial thrombosis or kinking 6, Arteriovenous thrombus 1	32 (3.6); venous 25, arterial 6, combined 1	32 (3.6, 16 within 24 h)	90.6	Na	3 (<0.1)	10 (1.1)	96.5	99.8
**Salgarello [35]**	Na	rSO_2_ ≤30% OR drop rate in rSO_2_ ≥ 20%	Na	0	0	Na	Na	0	0	Na	Na
**Steele [36]**	Co CE/hD, Ni NIRS	Co Na, Ni StO_2_ ≤40% OR drop rate StO_2_ ≥15%/h	Ni; Arterial thrombosis resulting in total loss 1 Hematomas with venous congestion/thrombosis 3 Vascular pedicle kinked during the inset and closure 2 Arterial spasm 1	Co 5 (9.4), Ni 7 (9.3)	Co 4 (7.5), Ni 3 (4)	Co 0, Ni 85.7	Co 14.5, Ni 10.7	Co 5 (9.4), Ni 1 (1.3)	0	Co Na, Ni 100	Co Na, Ni 100
**Vranken [27]**	CE/Doppler	Proposed; enlarged ΔStO_2_ ≥ 38%, decreased StO_2_ ≤ 43%	StO_2_ 43%; Second anastomosis 1, StO_2_ 44%; partial necrosis (day 5) 1	2 (6.9); venous congestion 1	2 (7)	100	5	0	1 (3.4)	Na	Na
**Whitaker [37]**	NIRS	StO_2_/THI ≤50% of starting value	Venous thrombosis requiring revision anastomosis 3, Minor debridement (after 3–5 days) 2, Evacuation hematoma; flap loss 1	4 (40); venous 3	3 (30)	75	6–13	1 (10)	0	100	100
**Yano [28]**	ICG/CE	Proposed; rSO_2_ < 55%	Subcutaneous hematoma (detachment anastomosis 3 weeks later) 1, Anastomosis revision; suspected inadequate venous drainage 1	venous 1 (4)	1 (4)	100	Na	0	0	Na	Na
**Calin [20]**	Na	Na	Na	0	0	Na	Na	0	0	Na	Na
**Kohler [29]**	HSI	Proposed; StO_2_ <40% and NIR <40	Venous thrombosis 4	venous 4 (18.2)	6 (27.3)	33.3	Nr 12 ± 6.6, partial 11.5 ± 2.1, Cr 30 ± 14.5	4 (18.2)	2 (9.1)	100	100
**Schulz (‘20) [38]**	Na	venous value change; THI 43% → 57%, StO_2_ 45 → 31%, NIR 43 → 25%, TWI 33 → 24%	Minor complication (e.g. wound edge necrosis) 15, Venous congestion radial flap resulting in loss 1	venous congestion 1 (6.3)	Na	0	Na	1 (6.3)	Na	Na	Na
**Schulz (‘21) [39]**	Na	Proposed; venous THI ≥53%, NIR ≤25%, TWI ≤43%, StO_2_ ≤22% arterial drop of StO_2_ ≤3%, THI ≤3%	Arterial embolism resulting in flap loss 1, partial flap necrosis caused by local impaired perfusion 9	arterial 1 (5.6)	1 (5.6, triple)	0	Na	1 (5.6)	9 (50)	Na	Na
**Thiem [22]**	HSI	Proposed; StO_2_ <45%, NIR <25%	Venous thrombosis 2, Arterial occlusion 1	3 (9.1); venous 2, arterial 1	3 (9.1, all FF)	33	Na	2 (6.1)	0	100	100

Co = control, Ni = NIRS, c = complication, Nr = no revision, Cr = complete revision, Na = not available, CE = clinical examination, hD = handheld Doppler, rSO_2_ = regional oxygen saturation, StO_2_ = hemoglobin oxygenation, NIR = Near-infrared Perfusion index, THI = tissue hemoglobin index, TWI = tissue water index. ∆ = Delta.

**Table 5 life-12-00065-t005:** Proposed warning values for vascular crisis, parameters indicative of vascular crisis and parameters to distinguish venous from arterial crisis using NIRS versus HSI.

Technique	Model	Proposed Warning Value	Vascular Crisis	Venous Congestion	Arterial Occlusion
NIRS	ViOptix [23] (ViOptix Inc., Fremont, Ca, USA) InSpectra [11] (Hutchinson Technology Inc., Hutchinson, Mn, USA)	rSO_2_ ≤ 30% OR drop rate in rSO_2_ ≥ 20% StO_2_ < 50% of initial value	HbO_2_, StO_2_ drop Hb rise	HbT rise	HbT drop
HSI	TIVITA [22,39] (Diaspective Vision GmbH, Am Salzhaff, Germany)	*Venous* THI ≥ 53%, NIR ≤ 25%, TWI ≤ 43%, StO_2_ ≤ 22% *Arterial* Drop of StO_2_ ≤ 3%, THI ≤ 3%	StO_2_, NIR low	THI high	THI low

NIRS = near-infrared spectroscopy, HIS = hyperspectral imaging, rSO_2_ = regional oxygen saturation, THI = tissue hemoglobin index, NIR = Near-infrared Perfusion index, TWI = tissue water index, StO_2_ = hemoglobin oxygenation, HbO_2_ = oxygenated hemoglobin, Hb = deoxygenated hemoglobin, HbT = total tissue hemoglobin concentration.

## 4. Discussion

Flap loss is a severe and feared complication after free tissue transfer in reconstructive microsurgery. Alongside the clinical assessment to detect signs of flap failure (either partial or total flap loss) in the early postoperative phase, objective monitoring of free flaps is expedient [11,40]. The ideal monitoring technique most importantly is objective, but also reliable, accurate, sensitive, continuous and user friendly, as defined by Creech and Miller [4]. NIRS and HSI are two different non-invasive monitoring methods that meet (almost) all criteria as described and have also proven to be suitable for detection of vascular compromise [9,23,37]. This study provides a systematic review in which a comparison between NIRS and HSI is made in detecting vascular compromise in reconstructive flap surgery compared to standard monitoring.

For NIRS, several commercial devices are available, such as FORE-SIGHT (Edwards Lifesciences, Irvin, CA, USA), INVOS (Medtronic, Minneapolis, MN, USA), EQUANOX^TM^ (Nonin Medical Inc., Plymouth, MN, USA) and ViOptix (ViOptix Inc., Fremont, Ca, USA). With these devices tissue oxygenation is measured continuously using non-invasive sensors, which need to be applied on the skin in the area of interest. Despite its proven added value in detection of vascular compromise, the technique is only implemented in 5% of the DIEP-flap procedures in clinical practice [8,9,10]. Recently, more research has been performed on implementing HSI to monitor flap viability after free flap surgery. Although data on the use of HSI in the clinical setting is scarce, several studies concluded HSI to be reliable and accurate [22,26,29,38]. In addition, in a recent study by Thiem et al. HSI showed to be able to detect malperfusion of flaps before clinical monitoring [41]. Measuring tissue oxygenation with this imaging modality is discontinuous but contactless: no sensors need to be applied on the skin. 

A lack of knowledge concerning the interpretation of values presented by the different devices could be an explanation for the low percentage in daily clinical use of NIRS measurements. Manufacturers use different algorithms to assess the tissue saturation values, apply different fixed ratios between arterial to venous blood volume and incorporate varying number and different wavelengths of near-infrared light [42]. Furthermore, they develop sensors with different transmitter-receiver spacing, resulting in different penetration depths, which also affects estimation and calculation of rStO_2_ [43]. Hence, it is difficult to define universal cut-off values necessitating prompt intervention [24,44]. In the included studies, most research was performed using the ViOptix device. For this particular device, Keller defined a threshold for rStO_2_ of an absolute value below 30% as predictive values for detection of vascular compromise [23]. For HSI, the diversity in used devices is currently limited. In all HSI observational studies included in this review, the Tivita system (Diaspective Vision GmbH, Am Salzhaff, Germany) was used for tissue oxygenation measurement. For this device no general cut-off values are defined yet, but most studies concluded a StO_2_ value below 30% to be an indication for circulatory compromise for which intervention would be recommended and justified [22,29,38,45]. Using continuous NIRS, measurement changes in tissue oxygenation can be monitored over time. A decrease of 20% from baseline for more than 60 minutes in duration is considered to be an indication for a lack in tissue perfusion. By HSI this continuity in monitoring is unfortunately not possible. Nevertheless, using HSI it has recently been shown feasible to detect circulatory compromise before standard clinical detection [41]. Therefore, both monitoring methods can be used to detect vascular compromise in the early postoperative period. 

Implementing NIRS as a monitoring tool is less labor intensive for the medical staff. Because measurements are continuous, only one member of the team needs to be trained in performing the measurements. When values decrease below a certain threshold, this member receives a text message stating an extra clinical examination of the flap needs to be performed [40,46]. For using HSI extra medical staff needs to be trained before using the device, because photos need to be taken on different time-points during the day in a standardized manner.

Since sensors need to be used to measure tissue oxygenation with NIRS, not all flaps can be monitored with NIRS. For example, when using the FORE-SIGHT system a flap dimension of at least 50 mm by 30 mm was necessary for proper sensor placement [24]. Furthermore, these sensors are not sterile. Therefore, measuring saturation can only be performed in the postoperative phase. These could also be reasons for the scarce implementation of NIRS in clinical practice. With HSI being a contactless measurement, all types of flaps (e.g., fascio-cutaneous, muscle, intestinal) can be included. For example, a probe fixation of NIRS for an intraoral flap is difficult, although the contactless measurement by HSI may be suitable for intraoral flap monitoring. On the other hand, a buried flap monitoring would be difficult by a contactless way. Furthermore, without applying sensors on the skin, the HSI technique is friendlier for the patient and more importantly, the measurements can also be performed during surgery.

With HSI four different parameters (StO_2_, rStO_2_, THI, TWI) are measured. When these parameters are combined it is possible to determine whether the observed changes in values are caused by an arterial inflow or a venous outflow track problem [18,29,38]. For monitoring free flaps this could be of added value. When using NIRS, this distinction can only be made with a few devices. For example, with the ViOptix, which is unavailable in Europe. Therefore, the number of available devices in this area are limited. 

The limited use of NIRS could also be due to the fact that implementing this technique comes with a price [9,26]. Implementing tissue oximetry costs $16,500 for the device and $150 per sensor according to Smit et al. In a different study, costs up to $30,000 for a device and $700–$1200 for a sensor are documented. Nevertheless, by implementing NIRS in standard protocol, vascular compromise could be detected in an early phase. Total flap loss could potentially be prevented and consequently duration of hospital stays shortened, resulting in a decrease of $1350–1700 per DIEP-flap procedure [33,34,47]. The costs for an HSI device are approximately $40,000 [38]. Initially implementing HSI would be more expensive than NIRS, but in the long term it could be more cost effective because no extra costs are required for buying the single use sensors. However, literature concerning cost effectiveness of HSI as a monitoring tool for flap viability is currently not available.

A limitation of the current literature study is the amount and quality of the included studies. In this review 21 studies were included. Sixteen reported on NIRS (n = 1970 patients) and five reported on HSI (n = 90 patients). All studies were observational cohort studies; accordingly, the average risk of bias was moderate. For this reason, randomized clinical trials with a larger patient population comparing the two monitoring techniques are mandatory. Moreover, defining solid cut-off values and performing an up-to-date cost-effectiveness evaluation regarding NIRS and HSI are required.

In conclusion, the authors believe that NIRS and HSI can have an added value in the detection of flap failure in the early postoperative phase. Both techniques have proven to be reliable, accurate and user-friendly monitoring methods, but do not (yet) replace the gold standard of clinical flap assessment. Based on the currently available literature, no firm conclusions can be drawn on which technique would be superior as an adjunct tool in free flap monitoring. 

## Figures and Tables

**Figure 1 life-12-00065-f001:**
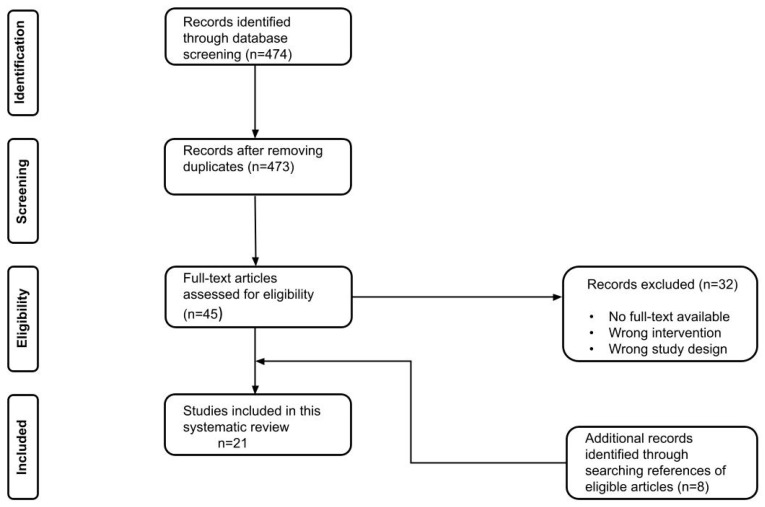
Flow chart of the included studies.

**Figure 2 life-12-00065-f002:**
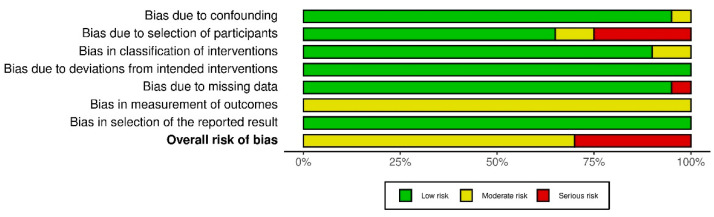
Summary of the risk of bias assessment of the included observational studies according to the ROBINS-I.

**Table 1 life-12-00065-t001:** Search strategy.

Category	MeSH Term	Free Search Term
#1: Population	Surgical flaps, or mastectomy, or perforator flap	Free flap OR Free tissue flap OR Surgical flaps OR Mastectomy OR Free tissue transfer flaps OR Perforator flap OR Mastectomy skin flap OR Mastectomy flap
#2: Intervention	Spectroscopy, near infrared, or hyperspectral imaging, or spectroscopies	Near infrared spectroscopy OR Noninvasive flap monitoring OR Flap monitoring OR Nirs OR Hyperspectral imaging OR Hsi OR Tissue oximetry OR Tivita tissue system OR Tivita OR Near infrared spectroscopies OR Near infrared spectrometry OR Near infrared spectrometries OR Spectrometries, near infrared OR Nir spectroscopies OR Nir spectroscopy
#3: Comparators	Venous insufficiency, or surgical wound dehiscence	Flap loss OR Partial flap loss OR Ischemia OR Necrosis OR Venous congestion OR Venous insufficiency OR Post operative complication OR surgical wound dehiscence

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
