# Peer review of "Near-Infrared Spectroscopy (NIRS) versus Hyperspectral Imaging (HSI) to Detect Flap Failure in Reconstructive Surgery: A Systematic Review"

_life, 2022, doi:10.3390/life12010065_

Round 1

Reviewer 1 Report

INTRODUCTION

The introduction concisely summarizes the intended research question with a solid consideration of the latest relevant literature.

MATERIAL AND METHODS including RESULTS

The methodological procedure for the literature search complies with the standard requirements for the preparation of a systematic literature review.

However, there is a major methodological issue according to the reviewer. The quality of any monitoring procedure depends on the correct-positive or false-negative detection rate of malperfusion. Overall, it is not clear on the basis of which monitoring method (control or HSI beziehungsweise NIRS) the decision to re-explore was based in the relevant studies. Thus, for example, the following sentence, "Salvage rate, the percentage of flaps with vascular crisis that 182 could be saved, overall was 81.1% (95% CI: 65.1-90.8). HSI salvage rate was 22.2% (95% 183 CI: 5.6-57.9) and NIRS salvage rate was 88.3% (95% CI: 80.1-93.4)." (page 5, line 182-184) as well as the results are of little to no significance, as it is not apparent on the basis of which method (control or HSI/NIRS) was re-explored (no indication of correct-positive / false-negative detection rate).

In addition, the number of cases with perfusion problems considered to be actually relevant for the assessment of a monitoring procedure is very small.

Regarding the time to detection of malperfusion using HSI compared to clinical control, a recent study is available ( a study by the reviewer himself) which showed for the first time the benefit of HSI in a collective of 19 re-explored flaps.

(Thiem DGE, Römer P, Blatt S, Al-Nawas B, Kämmerer PW. New Approach to the Old Challenge of Free Flap Monitoring-Hyperspectral Imaging Outperforms Clinical Assessment by Earlier Detection of Perfusion Failure. J Pers Med. 2021 Oct 27;11(11):1101. doi: 10.3390/jpm11111101. PMID: 34834453; PMCID: PMC8625540.)

DISCUSSION

Considering the above points, the discussion is well structured and addresses the most relevant points.

Author Response

The authors are grateful for the positive comments and are indebted to the reviewer for carefully reading their manuscript and providing detailed comments. Please see our reply below (changes in the text are underlined). The changes are highlighted as ‘tracked changes’ in the manuscript

Reviewer 2 Report

On the whole the article was written according to the PRISMA guideline, however, the main theme of this article “comparison of NIRS and HSI” was not clear and not easy to grasp.

Table 2,3,4 

The order of the studies is unstructured. At least the studies should be divided into two categories of NIRS and HSI.

Table 3 

The information of ‘Bilateral flap’ or ‘Weight’ seems unimportant. Rather the location of reconstruction is considered to be necessary. For example, an intraoral flap is difficult to be monitored by NIRS. Was there any study of HSI which dealt with intraoral flaps?   

Table 5

The table is about comparison of NIRS and HSI, however, the products were limited to just two (ViOptix and TIVITA). There are several products of NIRS and HSI, and each product has different measurable depth so that the proposed warning value differs one by one. This article is a systematic review, thus the information of all the products of the selected studies should be included in the table.

In addition, it is desirable that this table is converted to a comparative table of NIRS and HSI. For example, the probe of NIRS needs contact with the flap although the HSI has a contactless measurement system; the NIRS is of continuous measurement although the HSI is of intermittent measurement.     

Results

The statements about the comparison of NIRS and HSI are sufficient, however, how about making another table about the results of comparison (Flap survival, Vascular crisis, Salvage rate, Partial loss, etc.) to be more reader-friendly.

Discussion

“With HSI being a contactless measurement, all type of flaps (e.g. fascio-cutaneous, muscle, intestinal) can be included.”

In spite of the above statements, in the table 2, the studies of Guye and Yano applied the NIRS for gastro-omental flaps and free jejunum flaps, however, there is no intestinal flap in the HSI studies.   

A probe fixation of NIRS for an intraoral flap is difficult, although the contactless measurement by HSI may be suitable for intraoral flap monitoring. On the other hand, a buried flap monitoring would be difficult by a contactless way.

Can authors add any commentary about the above contents?

Author Response

(The authors gave the same response as above.)

Reviewer 3 Report

This paper compared the application value of NIRS and HIS in noninvasive tissue oxygen detection through systematic literature review. The blood support monitoring of free flap has certain guiding significance and clinical value. But there is some problem in this paper(the detail showed in attached word document). Please pay attention to and modify them.

Author Response

(The authors gave the same response as above.)

Round 2

Reviewer 1 Report

Most of the issues of concern have been adequately addressed. 

Reviewer 2 Report

The authors made appropriate changes to the manuscript except for a new table that shows the result overview of comparison between NIRS and HSI (Flap survival, vascular crisis, Salvage rate, Partial loss, etc.). The results are certainly written in the Results section, although readers usually see figures or tables at first and would need to make a little effort to find the results overview of the comparison.